# VILA² : VLM AUGMENTED VLM WITH SELF-IMPROVEMENT

## ABSTRACT

While visual language model (VLM) architectures and training infrastructures advance rapidly, data curation remains under-explored where quantity and quality become a bottleneck. Existing work either crawls extra Internet data with loose guarantee of quality or distills from black-box proprietary models (*e.g.*, GPT-4V / Gemini) that is API frequency and performance bounded. This work enables a VLM to improve itself via data enhancement, exploiting its generative nature. We introduce a simple yet effective VLM augmentation scheme that includes a self-augment step and a specialist-augment step to iteratively improve data quality and hence, model performance. In the self-augment step, the instruction-finetuned VLM recaptions its pretraining caption datasets and then retrains from scratch leveraging refined data. Without any expensive human-in-the-loop annotation, we observe improvements in data quality and downstream accuracy boosts with *three* self-augmentation rounds – a viable *free lunch* to current VLM training recipe. When self-augmentation saturates, we augment the caption diversity leveraging specialty skills picked up from instruction finetuning. We finetune VLM specialists from the self-augmented VLM with domain-specific experts, including spatial, grounding, and OCR, to fuse task-aware synthetic data into the pretraining stage. Data quality improvements and hallucination reductions are cross checked by VLM (GPT-4V, Gemini) and human judges. Combining self-augmentation and specialist-augmented training, VILA² consistently improves the accuracy on a wide range of benchmarks over prior art , producing a reusable pretraining dataset that is 300x more cost-efficient than human labeling. We will open source the code upon publication.

## 1 INTRODUCTION

The success of large language models (LLMs) (Raffel et al., 2020; Dai et al., 2019; Brown et al., 2020; OpenAI, 2023b; Touvron et al., 2023a;b; Taori et al., 2023; Chiang et al., 2023; Karamcheti et al., 2021; Chowdhery et al., 2022; yi, 2023; Bai et al., 2023a) has offered the cornerstone for cross-modality tasks. Through the alignment of visual encoders to LLMs, visual language models have enabled myriad appealing capabilities to visual tasks, such as instruction following, zero-shot generalization, few-shot in-context learning (ICL), and enhanced world knowledge (Liu et al., 2023c; Alayrac et al., 2022; Driess et al., 2023; Chen et al., 2023c; Li et al., 2023b; fuy, 2023; Bai et al., 2023b; OpenAI, 2023a; Zhu et al., 2023a; Lin et al., 2024). The field has progressed rapidly in the past two years, yielding effective alignment training recipes (Driess et al., 2023; OpenAI, 2023a; Lin et al., 2024) and model architectures (Liu et al., 2023c; Alayrac et al., 2022; Driess et al., 2023; Chen et al., 2023c; Li et al., 2023b).

Contrary to the fast-evolving training enhancement, the underlying human-generated datasets and tasks remain simple (Zhu et al., 2023b; Schuhmann et al., 2022; Byeon et al., 2022; Sharma et al., 2018). Given the costly nature of VLM training, most methods are confined with *coarse-quality large-scale* captioning image-text pairs (pretraining), followed by *fine-grained small-scale* supervised finetuning (SFT). Enhancement of image-text pairs with millions and billions of instances can inevitably impose a huge amount of human effort, and thus not realistic. Recent methods have observed rewarding distillation possibilities from proprietary commercial models like GPT-4V (OpenAI, 2023c) and Gemini (Gemini Team & other authors, 2023). However, the performance is upper bounded by

Figure 1: The schematic diagram to train VILA$^2$, short for VILA-augmented VILA. We re-formulate visual language model (VLM) training with *"model in the loop"* to remedy training data defficacy. We start with validating design options in constructing a self-augmenting loop (Section. 2.1) to improve on caption quality of the default training task. After the saturation of this process, we challenge the VLM to generate data conforming to extra SFT-enabled tasks to further VLM learning (Section. 2.2). Our new design insights yield off-the-shelf performance boosts to VLMs (Section. 3).

these models. In the meantime, studies remain very sparse on how to better utilize VLMs to correct human error and remedy dataset task simplicity for enhanced training.

In this work, we aim to answer "*whether it is possible that the VLM itself can remedy dataset deficiency and enhance its training.*" We delve deep into the potential of using VLM itself to refine and augment pretraining data and performance iteratively. Our new training regime, summarized in Figure 1, consists of two main steps: a **self-augment step** and a **specialist-augment step**. We start with the self-augment loop (Figure 1 (a)) that leverages VLMs to enhance the quality of pretraining data. We demonstrate that synthetic data, combined with the original data, can collaboratively generate stronger models in a bootstrapped loop manner. Intuitively and as we observed, the loops offer performance boosts *for free*, but suffers diminishing returns after **3** rounds. To facilitate further learning, we reformulate a more challenging task-specific loop (Figure 1 (b)). In this loop, specialists with a focus on new knowledge or tasks, such as a spatial-aware expert, OCR expert or grounding expert, are finetuned from the self-augmented VLM using a limited amount of additional SFT data. The specialists can then recaption a massive amount of pretraining data. Finally, the self-augmented VLM can be retrained on the specialist-recaptioned pretraining data to further boost the performance.

The insights yield a novel VLM augmentation training regime progressively improves data quality, by transferring knowledge from the **higher-quality but small-scale SFT** stage back to the **larger-scale but coarse pretraining** phase. This improvements cover enhanced visual semantics (Figure 1) and reducing hallucinations (Table 7- 8). We also observed consistent agreements on data quality improvements when cross checking the data by VLM models (GPT-4V, Gemini) and humans (Ph.D. students). This offers a direct performance boost to VLMs. We introduce a new family of VILA$^2$ models, as in VLM-augmented-VLM. VILA$^2$ outperforms state-of-the-art methods across main benchmarks, all enhanced without bells and whistles via self-bootstrapped training. We hope that the insights and release of VILA$^2$'s recipe, data, and code can assist with our community for better understanding and usage of synthetic data to train stronger VLMs.

## 2 METHODOLOGY

In this paper, we focus on auto-regressive VLMs where image tokens are projected into the textual space and concatenated with text tokens, in line with (Liu et al., 2023c; Gemini Team & other authors, 2023; Lin et al., 2024). This approach is chosen because of its flexibility when handling multi-modal inputs. We follow the widely adapted three-stage training paradigm, *i.e.*, align-pretrain-SFT, to ablate our studies. We start to self-augment VLM training by constructing a bootstrapped loop leveraging VLM's general captioning capability. After the bootstrapping saturates, we then introduce specialist augmenting exploiting VLM's skills picked up during SFT across additional visual tasks as specialist feedback to its pretraining stage. We next elaborate on these steps in detail.

### 2.1 SELF-AUGMENTING VIA GENERAL KNOWLEDGE ENHANCEMENT

Existing VLM training largely relies on data from the Internet, where the texts are usually brief and short, see Table 1 where the average number of words is less than 20 for MMC4 (Zhu et al., 2023b) and COYO (Byeon et al., 2022). In addition to brevity, human annotations can also fall short in explaining to LLMs the versatile semantics an image presents. As another example, Figure. 4 indicates that an original COYO caption that only describes a person riding on the street, omitting details about clothing and surroundings. Previous studies have either assigned humans to write dense

Figure 2: LLM judgement for captions from $VILA_i$, $i$ indicating the self-augmenting round. Evaluations are based on 5,000 sampled data from Coyo Byeon et al. (2022). Both GPT-4V OpenAI (2023c) and Gemini-1.0-pro Gemini Team & other authors (2023) prefer for $VILA^2$ augmented texts and captions from later rounds got higher scores.

| | MMC4 | COYO | COYO-$VILA_1$ | COYO-$VILA_2$ | COYO-$VILA_3$ | COYO-$VILA_4$ |
|---|---|---|---|---|---|---|
| Avg #Words | $17.1 \pm 25.0$ | $11.9 \pm 9.0$ | $101.2 \pm 43.0$ | $117.1 \pm 49.1$ | $126.77 \pm 50.10$ | $125.9 \pm 51.2$ |
| $VQA^{v2}$ | N.A. | 61.6 | 62.5 | 63.5 | **63.7** | 63.6 |

Table 1: The average number of words and $VQA^{v2}$ evaluation comparison between the original dataset and the re-captioned dataset. Best performance is bolded and second best is underlined. During self-augmentation, the caption lengths increases significantly, thus offering more details.

captions or by using commercial propriety APIs for detailed descriptions. The first option can be labor-intensive and costly, while the second risks model biases, limiting model performance, and potentially raising copyright concerns.

Rather than distilling proprietary models or relying on manual laboring, we aim to use *VILA to generate better captions for VILA's pretraining*. This approach exploits the power of the already-intelligent VILA within intermediate training stages to conduct laborious relabelling efforts. We begin with the original dataset to train the initial version of VILA, referred to as $VILA_0$ in subsequent experiments. Next, we use $VILA_0$ to re-caption VILA's pretraining datasets. With appropriate prompt choice and conversation template, $VILA_0$ is able to generate *long* and *detailed* captions. Then the augmented datasets, consisting of real images from the internet and synthetic texts from $VILA_0$, are used to train the next round of VILA, named $VILA_1$. This self-augmenting process can be repeated several rounds before convergence, leading to detailed descriptions (higher $VQA^{v2}$ score in Table 1) and improved text quality (LLM Judge in Figure. 2).

### 2.1.1 PROMPTS AND TEMPLATE DESIGN

The choice of prompt is particularly important for immediate performance improvements. To validate prompt design choices, we conducted an in-depth study on prompt choices as follows and discuss our findings, where `` indicates the location where image features will be inserted,

- Prompt Simple: ` Describe the image briefly.`

- Prompt Long-v1:` Describe the image in details.`

- Prompt Long-v2:` Elaborate on the visual and narrative elements of the image in detail.`

- Prompt Long-v3: ` Instead of describing the imaginary content, only describing the content one can determine confidently from the image. Do not describe the contents by itemizing them in list form. Minimize aesthetic descriptions as much as possible.`

**Brief and Short Re-captioning is NOT Helpful.** We begin with a straightforward prompt asking VLMs to *briefly* describe the image. Despite these brief recaptions being significantly longer than the original texts (90 vs. 17 words), there is no notable improvement in VLM benchmarks, as shown in Table 2. In fact, metrics even deteriorate in benchmarks such as Science-Image and MMMU-Test. This decline may stem from a lack of details during recaption.

Next, we redesign the prompt to encourage VLMs to provide a more detailed description of visual narrative elements in images. We also referenced the prompt template from ShareGPT-4V (Chen et al., 2023b) to ensure the descriptions are accurate and precise. Our experiments demonstrate that using three different long prompts improves the quality of recaptioning and boosts performance in

| | Avg #words | VQA$^{v2}$ | GQA | SQA$^I$ | VQA$^T$ | POPE | LLaVA$^W$ | MM-Vet | MMMU |
|---|---|---|---|---|---|---|---|---|---|
| Baseline | 17.1 | 79.6 | 62.4 | 68.4 | 61.6 | 84.2 | 68.4 | 34.5 | 33.8 |
| *Prompt Ablation for Self-Augmenting* | | | | | | | | | |
| Self-augment Iter1 - Simple | 90.4 | 79.4 | 63.0 | 68.7 | 62.4 | **87.0** | 68.3 | 34.5 | 33.1 |
| Self-augment Iter1 - Long v1 | 94.8 | 80.0 | 62.7 | **71.1** | 62.2 | 84.0 | 71.7 | 34.5 | 34.4 |
| Self-augment Iter1 - Long v2 | 105.4 | **80.1** | 63.2 | 70.7 | 62.7 | 84.6 | 71.7 | 34.9 | 34.7 |
| Self-augment Iter1 - Long v3 | 102.4 | **80.1** | 63.4 | 71.0 | 62.9 | 85.0 | 71.4 | 34.4 | 34.7 |
| *Conversation Template Ablation for Self-Augmenting* | | | | | | | | | |
| Mixed - re-caption text only | 101.2 | 79.6 | 62.5 | **71.1** | 62.3 | 81.0 | 71.8 | 34.2 | 34.1 |
| Mixed - concatenated | 127.3 | 80.0 | **63.2** | 71.0 | **62.5** | 85.0 | **72.2** | **34.8** | **35.8** |

Table 2: Comparison with different prompts and training templates when self-augmenting for one round. The best and second-best results are highlighted with **bold** and underline respectively. The results show that prompts design are critical for self-augmenting. Re-captioning the dataset with naive prompt "*Describe the image briefly*" does not improve while designed prompt can significantly boost the mode performance.

benchmarks, detailed in Table 1. Therefore, we leverage a mixture of these prompts by randomly selecting from versions v1 to v3.

**Keeping Original Human Text is Important.** We compare different conversation templates in Table 1. The first template uses only real human data, while the "concatenated" approach adapts both human and synthetic descriptions. Our experiments reveal that using self-augmented data improves performance on major benchmarks like LLaVA-Bench, Science-Image, TextVQA. However, we noticed a decline in several metrics. This prompted us to concatenate both the original and re-captioned texts to best preserve information, which consistently improves all VLM metrics. (Table 1).

## 2.2 SURPASSING THE LIMIT WITH SPECIALIST VLM AUGMENTATION

While self-augmentation provides a simple yet effective way to boost VLM's performance, we notice that the improvement starts to saturate with all free lunches having been squeezed (Table 3). We hypothesize that this shortcoming stems from the monotonic task of *general* descriptive captioning, which is also heavily influenced by language modeling priors.

To advance the boundary of self-augmentation, we propose the integration of extra *task-specific* knowledge into a generalist VLM to create several specialist VLMs. Each specialist model is finetuned with data that demands a deeper understanding of image components and semantics, *e.g.*, spatial relations, localization, and OCR. A bootstrapped loop can then transfer such specialist knowledge from small-scale SFT data onto a large number of pretraining images.

### 2.2.1 ACQUIRING SPECIALIZED KNOWLEDGE

We focus on three challenging tasks: *spatial relations understanding*, *grounded narration*, and *OCR* (Figure 3), and then elaborate the specialist construction as follows:

**Spatial Specialist.** To explicitly challenge the model to acquire additional spatial awareness, we curated *SpatialRelationQA*, a dataset containing 1 million conversations about spatial relations within scenes. Our dataset is built on LV3D, a comprehensive collection of both indoor and outdoor 3D datasets from Cube-LLM (Cho et al., 2024) that is designed to enhance the understanding of 3D spatial relations requiring both perceptual and grounding skills.

We formulated a two-step process to create the QA pairs.

1. For each cleaned 3D scene, we iterated through all 3D bounding boxes and randomly sampled from object-object relations (`closest, in front of, behind, left, right`) and object-camera relations (`close, far, closest, farthest, left, right`);

2. Next we checked if any remaining bounding boxes matched these sampled relations. For each matched results, we randomly selected question templates to construct the QA pairs, incorporating instances, their projected 2D bounding boxes, and relations.

A sample question can be constructed as: "*Where is the chair closest to the table [$x_{left}$,$y_{top}$,$x_{right}$,$y_{bottom}$] in the image?*", with answer being the target bounding box. During re-captioning, we guide the model to answer with more spatial information by prompting the specialist

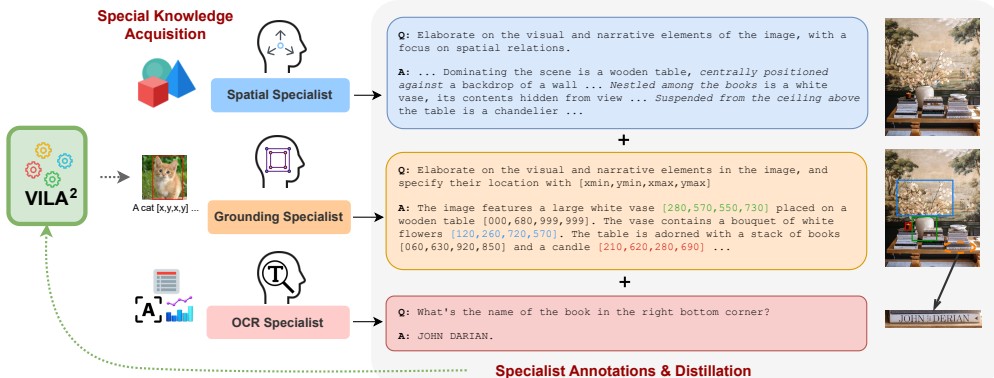

Figure 3: VILA$^2$ specialist VLM augmentation overview. We gather task-specific knowledge to create task-specialist VLMs. These specialist VLMs annotate images with task-oriented prompts and generate question-answering pairs to re-train the next iteration of VILA$^2$.

VLM with evenly sampled templates during the recaptioning phase, e.g., "*<image> Can you explain the content of the image and their spatial relations in detail?*".

**Grounding Specialist.** To enhance knowledge of grounding awareness, we exploited grounded narration – a highly visual-centric task that requires VLMs to generate detailed captions to accurately locate major visual elements using 2D bounding boxes, as shown in Figure 3. This approach provides dense supervision and allows us to verify if VLMs hallucinate. To develop the VILA$^2$ grounding specialist, we used image-grounded caption pairs from the 20M GRIT dataset (Peng et al., 2023). We first filtered out bounding boxes covering more than 70% of the image area, as many images in GRIT are book or album covers unrelated to the captions. We then removed images containing more than three instances of the same category to reduce complexity and decrease noise in generation orders. This process yielded 4M high-quality instances for grounding specialist training, which we split into two subsets: *Grounding-Short* (3M) and *Grounding-Long* (838K) for two-stage finetuning.

**OCR Specialist.** For OCR capabilities, we utilize a diverse set of images featuring textual content, such as tables, charts, and documents, to develop an OCR specialist. Each image is annotated with QA pairs that focus on text recognition (*e.g.*, *Q: What is the title of the book?*), comprehension (*e.g.*, *Q: Which bar has the largest value?*), and reasoning (*e.g.*, *Q: What is the main idea of the quote from Albert Camus?*). Dataset details are provided in appendix A.2.

The three specialist are then applied to the final augmentation stage. We use a new set of task-oriented prompts to activate the specialists' knowledge and improve their instruction-following ability by narrowing focus. Specifically, during the specialist augmentation stage. we prompt with evenly sampled templates of `"<image> Elaborate on the visual and narrative elements of the image in detail, with a focus on spatial relations."` and `"<image> Can you explain the content of the image and their spatial relations in detail?"` Similarly, the grounding specialist generates captions with bounding boxes for the major visual focus, while the OCR specialist identifies most textual content in the images. The responses from these different specialist VLMs are appended to the original captions as QA pairs for the next pretraining iteration of VILA$^2$. The full details are attached in appendix A.1.

## 3 EXPERIMENTS

**Model Architecture.** We follow the architecture from VILA (Lin et al., 2024), where a multi-modal large model consists of three key components: an *LLM* for auto-regressive generation, a *visual encoder* for extracting visual features, and an image-text *projector* to align the modalities.

We use Llama2-7B (Touvron et al., 2023b) for exploratory experiments to address the question "*To what extent can a VLM self-bootstrap?*". Then we switch to our previous SOTA settings with Llama3-8B-Instruct (Author, s) and Yi-34B (yi, 2023) when compared to other methods. For visual encoders, we use SigLIP (Zhai et al., 2023) for LLaMA-series models and InternViT-6B (Chen et al., 2023d) for the Yi-34B model. For projection layers, we follow LLaVA (Liu et al., 2023c; 2024c) to adapt simple linear layers for bridging image and text modalities. At the same time, we introduce a

**Raw caption**: Boris Johnson riding on the street.

**Self-Augmented Round-1**: In the image, a man is seen riding a bicycle on a city street. He is dressed in a blue suit and tie, giving him a professional appearance. The bicycle he's riding has a basket attached to it, suggesting that he might be using it for transportation or leisurely rides around the city. The background of the image reveals a lively urban scene. There are buildings visible, indicating that this is likely an urban area. A group of people can also be seen walking on the sidewalk, going about their day. Adding a touch of color to the scene is a yellow flag with red and black stripes hanging from one of the buildings. This could possibly indicate the presence of a business or organization associated with the building. Overall, the image captures a moment of everyday life in a bustling city, with the man on his bicycle adding a unique element to the scene.

**Self-Augmented Round-2**: The image captures a moment of surprise and excitement on the face of a man, who is riding a bicycle. He's dressed in a gray jacket and blue tie, giving him an air of professionalism. His blonde hair stands out against his dark jacket, adding to the contrasting elements in the scene. The background provides context to the setting. A brick building with a yellow awning suggests it could be a commercial area or perhaps a residential neighborhood. People can be seen walking by, going about their day, oblivious to the man's unexpected ride. A notable detail is the sign for "Bacardi" visible in the background. This could indicate that the location is near a Bacardi establishment, possibly a bar or restaurant. The sign adds an element of intrigue to the scene, hinting at stories untold beyond what we see.

**Self-Augmented Round-3**: The image captures a lively scene on a bustling street. At the center of the frame, a man is energetically pedaling a white bicycle. He's dressed in a black jacket and blue tie, adding a touch of formality to his otherwise casual attire. His mouth is open wide as if he's shouting or laughing, perhaps expressing excitement or surprise. The background of the image provides context to the location - it appears to be a busy city street with buildings and people going about their day. The photo seems to have been taken during the day under clear skies, casting bright light that illuminates the entire scene. Despite the hustle and bustle, there's a sense of joy and freedom conveyed by the man riding the bike, making this an interesting snapshot of city life.

**Specialist-Augmented**. In the image, a man is the central figure, riding a bicycle on a street. He is dressed in a gray jacket and a blue tie, giving him a professional appearance. The bicycle he's riding has a blue sign on the front, although the text on the sign is not discernible from the image. The man is not alone; he is accompanied by a group of people who are walking behind him. Their exact number is hard to determine from the image, but they appear to be a small crowd. The setting of the image is a street, with a building visible in the background. The building's details are not clear, but it adds context to the scene. The man on the bicycle is facing towards the right side of the image, suggesting that he might be moving in that direction. The people walking behind him are also facing the same direction, indicating that they might be following the man on the bicycle. Overall, the image captures a moment of everyday life, with the man on the bicycle as the main focus, and the people and the building providing the backdrop. The image does not contain any text. The relative positions of the objects suggest a dynamic scene, with the man on the bicycle leading the way for the people behind him.

Figure 4: VILA$^2$ continuously enhances caption quality over self-augmenting. The sample is from the COYO. We mark facts in green , hallucinations in red , and spatial information in blue . (Please zoom out for the best viewing experience)

$4\times$ downsampling of visual tokens by concatenating $2 \times 2$ neighboring patches along the channel dimension and using a simple MLP for the downsampling process.

**Training Strategies.** We conduct VILA$^2$ training following widely used three-stage settings.

1. *Projector Initialization*. The language models and ViT are separately pretrained, while the projector is randomly initialized. To initially align the feature space between the visual and text modalities, we utilize the LLaVA align dataset (Liu et al., 2023c).

2. *Vision-Language Pretraining*. We then pretrain the model (LLM and the projector) on the visual language corpus. We consider interleaved image text corpus (*e.g.*, MMC4 (Zhu et al., 2023b)) and image-text pairs (*e.g.*, COYO (Byeon et al., 2022)). **We apply our VILA$^2$ for the pretraining data** and the augmented data will be applied in this stage to replace original COYO captions.

3. *Visual Instruction-tuning*. Finally, we perform instruction tuning of the pretrained model on visual language instruction datasets. The blending details is attached in the appendix.

Without specifically mentioned, our experiments are conducted with 128 GPUs and a global batch size of 1024. We employ AdamW optimizer with learning rate $\{10^{-3}, 5 \times 10^{-5}, 2 \times 10^{-5}\}$ for aforementioned three stages respectively. Each stage is trained with one epoch with a 3% warmup strategy. No weight decay is applied. In some self-/specialist augmented training, VILA$^2$ may involve extra stage to further improve. Please refer to Section. 2.2 and Appendix A.5 for more details.

**Data**. Our pretraining stage consists of 6M sampled MMC4 (Zhu et al., 2023b), 25M sampled Coyo (Byeon et al., 2022), and the full ShareGPT4V (Chen et al., 2023b). To ensure a fair comparison, we only replace the text captions during our experiments while keeping all image sources the same. We use two SFT data blends for different purposes: a smaller blend of 1.8M samples for exploratory experiments in Table 3–Table 4; a larger blend of 5.9M samples augmented from VILA's training receipt, for SOTA experiments in Table 5–Table 6. Detailed SFT recipe and specialist data fulllist can be found in the Appendix. A.3-Appendix. A.4.

**Evaluation**. We ablate our models in the following common VLM benchmarks. Note that some metrics are shortened due to space limits. VQA$^{v2}$ (Goyal et al., 2017); GQA (Hudson & Manning, 2019); SQA: ScienceQA (Lu et al., 2022); VQA$^T$: TextVQA (Singh et al., 2019); POPE (Li et al.,

| | VQA$^{v2}$ | GQA | SQA$^I$ | VQA$^T$ | POPE | LLaVA$^W$ | MM-Vet | MMMU |
|---|---|---|---|---|---|---|---|---|
| VILA$_0$ - Baseline | 79.6 | 62.4 | 68.4 | 61.6 | 84.2 | 68.4 | 34.5 | 33.8 |
| VILA$_1$ | 80.0 | 63.2 | 71.0 | 62.5 | 84.6 | 72.2 | 34.8 | 35.8 |
| VILA$_2$ | 80.8 | **63.5** | 71.5 | 63.5 | 84.7 | 71.2 | 34.9 | 35.2 |
| VILA$_3$ | 80.7 | **63.5** | 71.5 | 63.7 | 84.5 | **72.3** | 35.5 | 35.5 |
| VILA$_4$ | 80.7 | 63.4 | 71.2 | 63.6 | 85.0 | **72.3** | 35.5 | 35.0 |
| VILA$_3$+Spatial Specialist | **81.1** | 62.8 | **72.9** | **65.0** | 85.0 | 71.4 | **37.1** | **36.8** |

Table 3: Self-augmenting can consistently enhance the performance of model training. For VILA$_{1-4}$, the best and second-best results are highlighted in **bold** and underline, respectively. With each iteration, VLM improves the quality of the pretraining dataset's captions. These improved descriptions lead to progressively better performance when training subsequent VLMs. Although the effects of self-augmentation plateau after three rounds, they can be further improved by our specialist.

2023c); MMB: MMBench (Liu et al., 2023d); MMB$^{CN}$: MMBench-Chinese (Liu et al., 2023d); SEED: SEED-Bench (Li et al., 2023a); LLaVA$^W$: LLaVA-Bench (In-the-Wild) (Liu et al., 2023c); MM-Vet (Yu et al., 2023); MMMU (Yue et al., 2024).

## 3.1 SELF-AUGMENTATION RESULTS

Our goal is to "augment" existing pretraining datasets by rewriting captions with dense and informative texts. Instead of relying on human labor or black-box APIs, we use VILA to generate these captions. Therefore, the enriched caption can help develop better VILAs based on which VILA can also feedback to further enhance the captions for the training dataset.

**VILA$^2$ Enriches Dataset Text Quality.** The main VLM's performance boost stems from improved caption quality. As shown in Table 1, caption length increases rapidly after self-augmentation and plateaus around rounds 3 and 4. This aligns with the trend observed in the benchmark results (round 1: 12 to 101, round 3: 117 to 126). Though Caption length does not increase significantly after round-1, we continue to observe consistent improvement on VLM benchmarks. We hypothesize that self-augmentation beyond round-1 primarily enhances caption quality by providing more accurate details and reducing hallucination, as visualized in Figure 4. The initial brief caption is brief and short (only describing Boris's riding action). The later captions evolves to include more details about clothing and surroundings. Although early iterations may contain some hallucinations (such as a non-existent basket and misread "Barcardi" text), subsequent iterations refine the caption to include only visual elements that can be confidently identified (more evidence in Table 7-Table 8).

**VILA$^2$ Bootstraps VLM's Performance.** We follow the same pretraining + SFT process as VILA (Lin et al., 2024) and sample 5% data from the pretraining phase to ablate. The images are from the existing COYO (Byeon et al., 2022) and MMC4 (Zhu et al., 2023b) and in each loop, we use the models trained last round to generate new captions for half of the sampled COYO images. MMC4 is not re-captioned because of its interleaved feature. Other settings are kept the same. We compare VILA$_i$ from different looping steps on common VLM benchmarks. We notice that self-augmented data help improves the model performance across different iterations: VILA$_{i+1}$ is consistently better than VILA$_i$ and the looping progressively boosts the performance (VILA$_{1-4}$ in Table 3). The self-augmenting technique is consistently useful until three rounds. VILA$_4$ reaches saturation and no longer bring consistent improvement of VILA$_3$.

| | VQA$^{v2}$ | GQA | VQA$^T$ | POPE | SEED-I | MME | MM-Vet | MMMU |
|---|---|---|---|---|---|---|---|---|
| *Pretrain Data: 10% MMC4-core+10% COYO-25M+ShareGPT4V-Pretrain* | | | | | | | | |
| Original Caption | 81.4 | 63.8 | 65.2 | 85.5 | 70.6 | 1472.5 | 34.0 | 31.8 |
| + Spatial Specialist | **81.9**$_{↑0.5}$ | **64.1**$_{↑0.3}$ | **66.0**$_{↑0.8}$ | 85.9$_{↑0.4}$ | 71.8$_{↑1.2}$ | 1476.5$_{↑4.0}$ | 36.7$_{↑2.7}$ | 32.5$_{↑0.7}$ |
| + OCR Specialist | 81.8$_{↑0.4}$ | 64.0$_{↑0.2}$ | 65.3$_{↑0.1}$ | 86.4$_{↑0.9}$ | **72.1**$_{↑1.5}$ | 1500.2$_{↑27.7}$ | 34.3$_{↑0.3}$ | 32.1$_{↑0.3}$ |
| + Grounding Specialist | 81.8$_{↑0.4}$ | 64.0$_{↑0.2}$ | 65.1$_{↓0.1}$ | **86.7**$_{↑1.2}$ | 71.0$_{↑0.4}$ | **1536.4**$_{↑63.9}$ | **37.5**$_{↑3.5}$ | **32.6**$_{↑0.8}$ |
| *Pretrain Data: MMC4-core+COYO-25M+ShareGPT4V-Pretrain* | | | | | | | | |
| Original Caption | 82.2 | 63.9 | 66.7 | **86.5** | 71.2 | 1518.2 | 42.6 | 33.4 |
| + All 3 Specialists | **83.0**$_{+0.8}$ | **64.7**$_{+0.8}$ | **70.9**$_{+4.2}$ | 86.4$_{-0.1}$ | **74.0**$_{+2.8}$ | **1656.2**$_{+142}$ | **44.7**$_{+2.1}$ | **35.8**$_{+2.4}$ |

Table 4: Effectiveness of the data re-captioned by specialists: We mark the best performance with **bold**. The results in the same block are trained with different pretraining data but the same SFT data. Specialists-annotated data consistently improves on both 10% and 100% pretraining setting.

| Method | LLM | Res. | VQA$^{v2}$ | GQA | VizWiz | SQA$^I$ | VQA$^T$ | MMB | MMB$^{CN}$ | SEED | LLaVA$^W$ | MM-Vet |
|---|---|---|---|---|---|---|---|---|---|---|---|---|
| BLIP-2 (Li et al., 2023b) | Vicuna-13B | 224 | 41.0 | 41 | 19.6 | 61 | 42.5 | – | – | 46.4 | 38.1 | 22.4 |
| InstructBLIP (Dai et al., 2023) | Vicuna-7B | 224 | – | 49.2 | 34.5 | 60.5 | 50.1 | 36 | 23.7 | 53.4 | 60.9 | 26.2 |
| InstructBLIP (Dai et al., 2023) | Vicuna-13B | 224 | – | 49.5 | 33.4 | 63.1 | 50.7 | – | – | – | 58.2 | 25.6 |
| Qwen-VL (Bai et al., 2023b) | Qwen-7B | 448 | 78.8 | 59.3 | 35.2 | 67.1 | 63.8 | 38.2 | 7.4 | 56.3 | – | – |
| Qwen-VL-Chat (Bai et al., 2023b) | Qwen-7B | 448 | 78.2 | 57.5 | 38.9 | 68.2 | 61.5 | 60.6 | 56.7 | 58.2 | – | – |
| LLaVA-1.5 (Liu et al., 2023b) | Vicuna-1.5-7B | 336 | 78.5 | 62.0 | 50.0 | 66.8 | 58.2 | 64.3 | 58.3 | 58.6 | 63.4 | 30.5 |
| LLaVA-1.5 (Liu et al., 2023b) | Vicuna-1.5-13B | 336 | 80.0 | 63.3 | 53.6 | 71.6 | 61.3 | 67.7 | 63.6 | 61.6 | 70.7 | 35.4 |
| VILA-7B (Lin et al., 2024) | Llama 2-7B | 336 | 79.9 | 62.3 | 57.8 | 68.2 | 64.4 | 68.9 | 61.7 | 61.1 | 69.7 | 34.9 |
| VILA-13B (Lin et al., 2024) | Llama 2-13B | 336 | 80.8 | 63.3 | 60.6 | 73.7 | 66.6 | 70.3 | 64.3 | 62.8 | 73.0 | 38.8 |
| LLaVA-NeXT-8B (Liu et al., 2024c) | Llama 3-8B | 672 | – | 65.2 | – | 72.8 | 64.6 | 72.1 | – | – | 80.1 | – |
| Cambrian-1-8B (Tong et al., 2024) | Llama 3-8B | 1024 | – | 64.6 | – | 80.4 | 71.7 | 75.9 | – | – | – | – |
| Mini-Gemini-HD-8B (Li et al., 2024b) | Llama 3-8B | 1536 | – | 64.5 | – | 75.1 | 70.2 | 72.7 | – | – | – | – |
| VILA$^2$-8B (ours) | Llama 3-8B | 384 | 82.9 | 64.1 | 64.3 | 87.6 | 73.4 | 76.6 | 71.7 | 66.1 | 86.6 | 50.0 |

Table 5: Comparison with state-of-the-art methods on 10 visual-language benchmarks. Our models consistently improve VILA under a head-to-head comparison, showing the effectiveness of enhanced pretraining data quality. We mark the best performance **bold** and the second-best underlined.

## 3.2 SPECIALIST AUGMENTATION RESULTS

The "self-augmentation then training" cycle reaches a plateau after three iterations, as illustrated in Table 3. However, by incorporating tasks-specific specialists, we can overcome the limit and introduce further performance improvement.

**Surpassing the Limit with VILA$^2$ Specialist.** The caption augmented by the specialist (the last example) retains the most visible details and provides more information about spatial relations compared to the "Round-4" caption (Figure 4). This additional information includes object-to-object relations, as well as localization and clear pose of the major focus, which are not present in the *SpatialRelationQA* dataset. We hypothesize this improvement might result from combining knowledge in specialist data and VLM's training data. The effectiveness of these enriched captions is demonstrated in Table 3 (VILA$_3$ + *spatial specialist*). Following the same SFT stage, we observe notable improvements in 5 out of the 8 benchmarks.

**The More Specialists, The Better The Performance.** We next explore the significance of scaling up our VILA$^2$ specialists with more pretraining data using recent state-of-the-art settings. We demonstrate the effectiveness of each specialist using S2 (Shi et al., 2024) with Llama 3-8B-Instruct (meta, 2024). On a 10% subset of pretraining data, specialist VLMs show overall improvements across most VQA benchmarks in Table 4's first part. We then combine annotations from all three specialists into multi-round QA pairs for each image and retrain VILA. This synergy among the specialists proves highly effective, with scaling up to the full pretraining set yielding significant improvements. Results on larger models will be discussed in next section.

## 3.3 BENCHMARK COMPARISON TO PRIOR WORK

We now perform a comprehensive comparison to prior work over 10 major benchmarks and summarize results in Table 5 and Table 6. Note that we used a total of 25 million COYO data sampled from the 700 million with the highest CLIP score. We augment the original short real labels with multi-round QA pairs annotated by three specialists, all from 8B models. For 40B models, we continue to train from the stage 2 checkpoints with a mix of 3.75 M recaptioned COYO and a 200K caption dataset (Chen et al., 2024). We observed the improvements in quality is consistent and can scale to 40B VILA checkpoints. The detailed training recipes of our 8B and 40B checkpoints are included in the Appendix and will be released jointly with the code base.

Remarkably we observed the enhanced self-augmentation and specialist augmentation training recipes, backed by enhanced and refined datasets, support VILA$^2$ to further push the performance boundary of VILA (Lin et al., 2024) by noticeable margins across almost all benchmarks, consistent with the ablated performance boosts we observed in previous analysis of Table 3. Moreover, VILA$^2$ now constitutes a SOTA performance on the main MMMU (Yue et al., 2024) test dataset leaderboard across all open-sourced models, without the usage of proprietary datasets and only based on publicly available datasets.

## 3.4 GAUGING ON SYNTHETIC DATA QUALITY AND HALLUCINATION

Figure 2 presents cross-evaluation results with Gemini and GPT-4V, showing their increased preference for captions generated in later rounds of self-augmentation. We also provide evidence from an

| Method | Overall (Test/Val) | Art & Design | Business | Science | Health | Human & Social | Tech. & Eng. |
|---|---|---|---|---|---|---|---|
| GPT-4V (OpenAI, 2023a) | 56.1 / 56.8 | 65.3 | 64.3 | 48.4 | 63.5 | 76.3 | 41.7 |
| SenseChat-V (Sensetime, 2024) | 50.3 / 54.6 | 62.7 | 44.1 | 42.3 | 55.7 | 74.7 | 43.5 |
| **VILA$^2$-40B (ours)** | **47.9 / 53.0** | **62.0** | **42.3** | **38.5** | **51.9** | **71.9** | **42.3** |
| Qwen-VL-MAX (Bai et al., 2023a) | 46.8 / 51.4 | 64.2 | 39.8 | 36.3 | 52.5 | 70.4 | 40.7 |
| InternVL-Chat-V1.2 (Chen et al., 2023d) | 46.2 / 51.6 | 62.5 | 37.6 | 37.9 | 49.7 | 70.1 | 40.8 |
| Cambrian-1-34B (Liu et al., 2024c) | - / 49.7 | - | - | - | - | - | - |
| LLaVA-1.6 (Liu et al., 2024c) | 44.7 / 48.1 | 58.6 | 39.9 | 36.0 | 51.2 | 70.2 | 36.3 |
| Mini-Gemini-HD-34B (Liu et al., 2024c) | - / 48.0 | - | - | - | - | - | - |
| Marco-VL-Plus* | 44.3 / 46.2 | 57.4 | 34.7 | 38.5 | 48.7 | 72.2 | 36.7 |
| Yi-VL (yi, 2023) | 41.6 / 45.9 | 56.1 | 33.3 | 32.9 | 45.9 | 66.5 | 36.0 |
| Qwen-VL-PLUS (Bai et al., 2023a) | 40.8 / 45.2 | 59.9 | 34.5 | 32.8 | 43.7 | 65.5 | 32.9 |
| Marco-VL-Plus* | 40.4 / 41.2 | 56.5 | 31.0 | 31.0 | 46.9 | 66.5 | 33.8 |
| Weitu-VL-1.0* | 38.4 / - | 56.6 | 30.5 | 31.1 | 38.4 | 59.0 | 34.2 |
| **VILA$^2$-8B (ours)** | **38.3 / 40.8** | **54.3** | **32.0** | **29.3** | **39.7** | **56.8** | **34.4** |
| InfiMM-Zephyr (Team, 2024) | 35.5 / 39.4 | 50.0 | 29.6 | 28.2 | 37.5 | 54.6 | 31.1 |
| SVIT (Zhao et al., 2023) | 34.1 / 38.0 | 48.9 | 28.0 | 26.8 | 35.5 | 50.9 | 28.8 |
| Emu2-Chat (Sun et al., 2023) | 34.1 / 36.3 | 50.6 | 27.7 | 28.0 | 32.4 | 50.3 | 31.3 |
| BLIP-2 FLAN-T5-XXL (Li et al., 2023b) | 34.0 / 35.4 | 49.2 | 28.6 | 27.3 | 33.7 | 51.5 | 30.4 |
| InstructBLIP-T5-XXL (Dai et al., 2023) | 33.8 / 35.7 | 48.5 | 30.6 | 27.6 | 33.6 | 49.8 | 29.4 |
| LLaVA-1.5 (Liu et al., 2023c) | 33.6 / 36.4 | 49.8 | 28.2 | 25.9 | 34.9 | 54.7 | 28.3 |

Table 6: Comparison with state-of-the-art methods on the MMMU dataset. *: model on leaderboard with unidentified reference. The models are ranked by overall test set scores (we report scores in a test/validation manner), including both proprietary and open-sourced models. We highlight our results with color green. VILA$^2$ achieves SOTA performance in the open source category.

out-of-distribution benchmark and a human blend quality ranking, demonstrating that hallucinations do not increase in these later rounds.

**Left-out Benchmark Results.** We first select the Visual Spatial Reasoning (VSR (Liu et al., 2023a)) benchmark which does not appear in our training set. This benchmark consists of triplets containing an image, a spatial-focused expression, (*e.g.*, *the cow is ahead of the person*), and a *True* or *False* label indicating its correctness. We observed reduced hallucinations on the VSR benchmark with more iterations of self-augmentation and have not yet reached a plateau, results are shown in Table 7.

**Human Judge Ranking Test.** We further add a more rigorous human test that compares the (re-)captions of 200 randomly sampled images from 11 human evaluators (most are PhD students). These evaluators were tasked with determining which caption exhibits fewer hallucinations, without any knowledge of the sources of the captions. We calculated the win rates of the captions from later self-augmentation rounds against those from earlier rounds, as detailed in Table 8. Human preference reflects a decrease in hallucinations with each additional round of self-augmentation, reaching saturation at Round-3. This observation aligns with the performance trends noted in Table 3 and GPT-4V and Gemini judges in Figure 2.

| Model | VSR *random* (%) ↑ | VSR *zero-shot* (%) ↑ |
|---|---|---|
| VILA$_0$ | 73.5 | 63.1 |
| VILA$_1$ | 75.1 | 64.8 |
| VILA$_2$ | 76.8 | 65.8 |
| VILA$_3$ | **77.4** | **66.4** |

| Human Evaluation | Win Rate (%) ↑ |
|---|---|
| Later Rounds vs Original Caption | 71.6 |
| Later Rounds vs Round-1 | 54.6 |
| Later Rounds vs Round-2 | **55.6** |
| Later Round vs Round-3 | 48.3 |

Table 7: Accuracies on the VSR benchmar.  Table 8: Preferences for 200 images' captions.

## 3.5 IMPROVED DATA QUALITY MATTERS MORE THAN INCREASED COMPUTATION

We next present additional ablations of VILA$^2$'s self-augmentation loop in comparison to training additional epochs on the same data, with results shown in Table 9. We can observe that simply scaling up epochs with coarse image/text pairs yields no performance boosts, despite the extra training costs. In contrast, enhancing the quality of data, as demonstrated in VILA, provides a more rewarding path.

| Model Variation | GQA | SQA$^I$ | VQA$^T$ | POPE | MM-Vet | MMMU |
|---|---|---|---|---|---|---|
| VILA$_0$-*Baseline* | 62.4 | 68.4 | 61.6 | 84.2 | 34.5 | 33.8 |
| Train one extra epoch | 62.5 | 68.7 | 61.9 | 84.0 | 34.4 | 33.9 |
| VILA$_1$ | 63.2 | 71.0 | 62.5 | 84.6 | 34.8 | 35.8 |
| Train two extra epochs | 62.3 | 69.0 | 61.7 | 83.9 | 34.4 | 33.7 |
| VILA$_2$ | 63.5 | 71.5 | 63.5 | 84.7 | 35.5 | 35.2 |

Table 9: Comparison between training additional epochs *on the same data* and training additional epochs *with self-augmentation*. Models do not benefit from more computations on identical data.

| Method | LLM | VT | # TPI | PT | VQA$^{v2}$ | SQA$^I$ | VQA$^T$ | MMB | SEED | LLaVA$^W$ | MM-Vet | MMMU |
|---|---|---|---|---|---|---|---|---|---|---|---|---|
| MM1-7B-Chat (Mar. 2024) | 7B | 300M | 720 | >2B | 82.8 | 72.6 | 72.8 | 72.3 | 64.0 | 81.5 | 42.1 | 35.6 |
| Idefics2-8B (Apr. 2024) | 8B | 400M | 320 | >600M | 81.2 | – | 73.0 | **76.7** | – | – | – | 37.7 |
| VILA$^2$-8B (ours, May 2024) | 8B | 400M | 196 | **51M** | 82.9 | 87.6 | 73.4 | 76.6 | 66.1 | 86.6 | 50.0 | 38.3 |

Table 10: Comparison of multimodal methods across benchmarks, with different settings of large language model parameters (LLM), vision tower parameters (VT), number of tokens per image (# TPI), and pre-training data size (PT).

## 3.6 Efficiency and Effectiveness of VILA$^2$

**Cost Analysis – Labeling.** Data quantity and quality are critical factors in model training. While VILA$^2$ involves three rounds of recaptioning, it is still far more cost-efficient than traditional human re-labeling. For example, a standard re-labeling on Amazon Turk costs 36 USD per 1k images, while VILA$^2$ costs only 0.12 USD per 1k images. The cost breakdown includes AWS pricing for H100 GPUs: USD 4.91 per hour for one H100, or USD 39.33 per hour for eight. With an inference speed of 10.6 images per second per H100, VILA$^2$ processes around 38,340 images per hour, making it **300x** cheaper and significantly faster.

**Cost Analysis – Training.** As shown in Table 10, VILA$^2$ achieves better accuracy with significantly less data (51M) compared to other works like MM1 (>2B) and Idefics2 (>600M). Even with three rounds of iteration, VILA$^2$ remains more cost-effective in terms of training computation. Further, VILA$^2$ is a *one-time cost* that can be leveraged for training multiple models. Once recaptioning is complete, this data can be shared with the community, reducing the need for expensive data pipelines.

## 4 Related Work & Limitations

**Visual language models (VLM).** Visual language models have rapidly progressed in recent years (Radford et al., 2021; Li et al., 2022a; Dai et al., 2023; Liu et al., 2023c; Ye et al., 2024; Cheng et al., 2024). The success mainly comes from pretraining visual and language models on the internet-scale data. Kosmos-2 (Peng et al., 2023) and PaLI-X (Chen et al., 2023c) largely scaled the pretraining data by pseudo-labeling bounding boxes from performant open-vocabulary object detectors (GLIP (Li et al., 2022b) and OWL-v2 (Minderer et al., 2024), respectively). They examined that strong perception capabilities such as object detection and OCR translate to better high-level reasoning tasks like visual question-answering (VQA).

**Contributions & Novelty.** Our work expands the horizon of data-scaling through our self-augmenting paradigm. ShareGPT4V (Chen et al., 2023b) applied a single round of recaptioning by distilling from GPT-4V. In contrast, we focus on a more general approach of *using VLM to augment VLM itself* without relying on commercial APIs or distilling from larger models. We provide 1) a detailed analysis of self-augmentation, covering prompt templates, iteration rounds, saturation points; 2) a practical method that uses specialist-augmentation to continually improve; 3) curated datasets that can be reused for future research. Our solution efficiently enhances SOTA VLM performance without requiring extra data or expensive APIs for closed-source models.

**Limitations**. Due to resource constraints, we concentrate on the design of a self-augmented data curation pipeline and verify the 7B, 8B, and 40B models with 51M data. Larger models (e.g., 70B and 405B) and more data (>0.5B pretrain data) can have the potential to lead to better VLM capabilities with self-augmenting abilities. We will address these aspects in future work.

## 5 Conclusions

This work has explored the techniques, insights, and benefits of using VLMs to self-improve its pre-training. We introduced two primary augmentation loops, one leveraging VLM's general captioning capacities and the other harnessing their strength in specialized visual tasks. We demonstrated the feasibility of three 'free lunch' rounds for VLMs through self-bootstrapping, with further improvements via knowledge distillation from specialist VLMs. Our new VILA$^2$ models demonstrate SOTA performances across a comprehensive set of benchmarks. Fruitful future directions include a deeper delve into the potential synergy between synthetic and real data to train stronger foundation models.

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

# A APPENDIX / SUPPLEMENTAL MATERIAL

## A.1 PROMPTS FOR SPECIALIST-AUGMENTATION

We use the following prompts during specialist- augmentation,

- Spatial Relations Understanding Specialist

```
"<image> Elaborate on the visual and narrative elements of the
image in detail, with a focus on spatial relations."
```

- Grounded Narration Specialist

```
"<image> Elaborate on the visual and narrative elements in the
image, and specify their location with [xmin,ymin,xmax,ymax]."
```

- OCR Specialist

```
"<image> Your task is to recognize and describe the text in the
image.  Please provide a brief description that focuses on the
textual content."
```

## A.2 SPECIALIST ACQUISITION

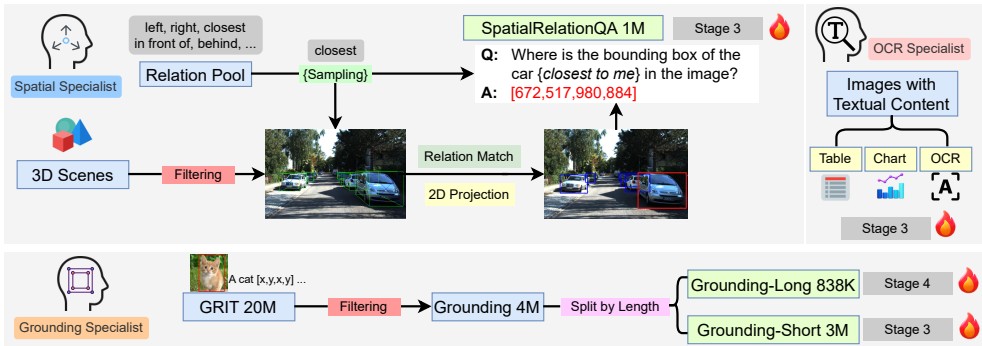

Figure 5: VILA[2] Specialist VLM Acquisition Pipeline. We gather task-specific knowledge from public datasets, followed by filtering noisy samples using rule-based strategies. We then train the specialist VLMs from pretrained checkpoints, employing different data blends and training strategies.

Specialty and expertise can be obtained via gathering existing data from the open-source community, human labeling, and annotating by domain-specific models, e.g. OCR models for text recognition and detection models for bounding box prediction. We also experimented with using open-world detectors like OWLv2(Minderer et al., 2024) to automatically label bounding boxes, VLMs to generate detailed captions, and LLMs such as Llama3-70B-Instruct to merge the information for the grounded narration specialist. However, we found that language models introduced more hallucinations into the merged grounded narration. This is because many different detection labels can share the same meaning and refer to the same instance, making it difficult for the language model to perform the bipartite matching between bounding boxes and their text correspondence.

## A.3 SFT DATA

We use two different datasets for our experiments: a 1.8M sample dataset for exploratory experiments and a 5.9M sample dataset for state-of-the-art experiments.

- **1.8M SFT Blend:** This dataset includes samples from the following sources: LLaVA-SFT, MSR-VTT, TextCaps, Image Paragraph Captioning, CLEVR, NLVR, VisualMRC, ActivityNet-QA, iVQA, MSRVTT-QA, MSVD-QA, DVQA, OCRVQA, ST-VQA, ViQuAE, VQAv2-train, Visual Dialog, GQA-train, FLAN-1M.
- **5.9M SFT Blend:** This dataset comprises all the datasets listed in the following table:

| Categories | Datasets |
|---|---|
| Hybrid | LLaVA-SFT, The Cauldron (subset) |
| Captioning | MSR-VTT (Xu et al., 2016), TextCaps, LLaVAR, Image Paragraph Captioning (Krause et al., 2017), ShareGPT4V-100K |
| Reasoning | CLEVR (Johnson et al., 2017), NLVR, VisualMRC (Tanaka et al., 2021) |
| Multi-images | ActivityNet-QA (Yu et al., 2019), VQAv2-train, iVQA (Yang et al., 2021), MSRVTT-QA, STEM-QA (Shen et al., 2024) |
| OCR | DVQA, OCRVQA, ST-VQA (Biten et al., 2019), SynthDoG-en, TextOCR-GPT4V, ArxivQA |
| World Knowledge | WIT (Srinivasan et al., 2021) |
| General VQA | ScienceQA-train, VQAv2-train, ViQuAE (Lerner et al., 2022), Visual Dialog (Das et al., 2017), GQA-train (Hudson & Manning, 2019), SHERLOCK (Hessel et al., 2022), Geo170K (Gao et al., 2023), MMC-Instruction (Liu et al., 2024b), LRV-Instruction (Liu et al., 2024a), RefCOCO-train (Yu et al., 2016) |
| Text-only | FLAN-1M (Wei et al., 2022), MathInstruct (Yue et al., 2023), Dolly (Conover et al., 2023), GSM8K-ScRel-SFT (Yuan et al., 2023) |

Table 11: Data mixture for the SFT stage.

### A.4 SPECIALIST DATA

We integrated specialty data with high-quality image captioning datasets and diverse instruction finetuning datasets, ensuring the models retain their narrative and instruction-following abilities while acquiring task-specific knowledge.

1. **Spatial Specialists.** We continued training the specialist from the stage 2, ALLaVA caption (Chen et al., 2024), and GPT-4V caption from ShareGPT4V (Chen et al., 2023b).

2. **Grounding Specialist.** We split the cleaned 4M grounded narration into *Grounding-Short* 3M and *Grounding-Long* 838K for a two-stage training process. In stage 3, we combined *Grounding-Short* 3M with ALLaVA caption (Chen et al., 2024) to adapt to new tasks of grounded narration while maintaining the narrative ability. In stage 4, we combine *Grounding-Long* 838K with Shikra GPT-4 (Chen et al., 2023a), Visual7W (Zhu et al., 2016), LLaVA-SFT, and 100K GPT-4V captions from ShareGPT4V to sustain both narrative and instruction following capacities.

3. **OCR Specialist.** We trained our OCR specialist with various internet datasets focused on text recognition, understanding, and reasoning, including LLaVA-SFT, TextOCR-GPT4V (Carter, 2024), SynthDoG-En (Kim et al., 2022), OCRVQA (Anand et al., 2019), TextCaps (Sidorov et al., 2020), ArxivQA (Li et al., 2024a), DocVQA (Kafle et al., 2018), AI2D (Kembhavi et al., 2016), ChartQA (Masry et al., 2022), LLaVAR (Zhang et al., 2023) and 35 OCR-related datasets from The Cauldron (Laurençon et al., 2024).

### A.5 TRAINING DETAIL

We adjust our training strategies akin to varying language model sizes for training cost considerations. We next elaborate on the details.

#### A.5.1 7B & 8B & 13B MODELS

We divide the entire training process of 7B&8B&13B models into three sub-stages.

- **Stage 1: Alignment Stage**. We train only the multi-modal projector using 595K image-text pairs, as mentioned in LLaVA, to achieve the initial alignment between the two modalities.
- **Stage 2: Pretraining Stage**. We gather a total of 51 million images, consisting of 25 million image-text pairs with the highest CLIP scores from COYO-700M, 25 million images

| Method | LLM | Res. | VQA$^{v2}$ | GQA | VizWiz | SQA$^I$ | VQA$^T$ | MMB | MMB$^{CN}$ | SEED | LLaVA$^W$ | MM-Vet |
|---|---|---|---|---|---|---|---|---|---|---|---|---|
| VILA$^2$-8B (ours) | Llama 3-8B | 384 | 82.9 | 64.1 | 64.3 | 87.6 | 73.4 | 76.6 | 71.7 | 66.1 | 86.6 | 50.0 |
| VILA$^2$-40B (ours | Yi-34B | 448 | 85.1 | 64.7 | 62.2 | 93.2 | 75.9 | 83.9 | 82.9 | 77.0 | 93.6 | 53.4 |

Table 12: Improvements from 8B to 40B checkpoints on 10 visual-language benchmarks.

in an interleaved image-text format from the MMC4-Core subset, and 1 million images with detailed captions from ShareGPT4V-Pretrain. During this stage, we unfreeze both the multi-modal projector and the language model to enhance comprehension of the diverse multi-modal training data. **We use the augmented data here to replace the original COYO captions.**

- **Stage 3: Supervised Finetuning Stage**. After stage 2, we collect diverse visual question-answer pairs and unfreeze all parameters to finetune the model for general-purpose VQA capacities.

### A.5.2 40B MODEL

For the VILA$^2$-40B model, we skip the cost-intensive stage 2 and train the model with 7.5 million images randomly sampled from the 25 million COYO subset pairing with various caption sources: 2.5 million with original COYO captions, 2.5 million with VILA$_3$ re-captioned descriptions, and 2.5 million with VILA$_3$ spatial specialist re-captioned descriptions. Both the multi-modal projector and the language model remain unfrozen. Note that adding interleaved data, such as MMC4, can further boost the performance and we leave this potential to future work. A detailed profiling of 40B performances over benchmarks is also included as Table 12.

### A.6 HYPERPARAMETERS

We use a universal batch size of 1024, a cosine decay learning rate schedule, a 0.03 learning rate warmup ratio, no weight decay, and AdamW as the optimizer for stable training, and details are expanded in Table 13. All trainings are conducted with 128 A100 GPUs.

| Hyperparameter | Stage 1 | Stage 2 | Stage 3 | Stage 4 |
|---|---|---|---|---|
| batch size | 1024 | 1024 | 1024 | 1024 |
| learning rate (lr) | 1e-3 | 5e-5 | 2e-5 | 1e-4 |
| lr schedule | cosine | cosine | cosine | cosine |
| lr warmup ratio | 0.03 | 0.03 | 0.03 | 0.03 |
| weight decay | 0 | 0 | 0 | 0 |
| epoch | 1 | 1 | 1 | 1 |
| optimizer | AdamW | AdamW | AdamW | AdamW |
| DeepSpeed stage | stage2 | stage3 | stage3 | stage3 |

Table 13: The detailed training setup for VILA$^2$ and the hyper-parameters across the training stages.

### A.7 ADDITIONAL RE-CAPTION COMPARISONS

We provide additional VILA$^2$ recaptioned examples from the SAM dataset comparing the baseline captions from alternative methods, *e.g.*, the widely adopted InstructBLIP Flan-T5 XXL (Dai et al., 2023) and LLaVA-NeXT-34B (Liu et al., 2024c), with captions generated through various rounds of VILA$^2$'s intermediate models in Figure 6 through Figure 8. The correct facts are marked with green, hallucinations are marked with red and spatial related information are marked with blue. From these examples, we can see that while the baseline caption is descriptive, it contains more hallucinations. As the re-captioning and training cycle progresses, the model tends to generate captions with more details and fewer hallucinations. Additionally, the spatial specialist introduces more spatial information into the recaptions, as expected.

**InstructBLIP**: The image depicts a train traveling along a hillside near a stone tower, which is visible in the background of the image. The train appears to be moving at a steady pace, and it can be seen passing by the tower several times during its journey. There are several people present in the scene, including a man standing next to the train, a woman sitting on the side of the train, and a young boy walking behind the train. All of these individuals appear to be enjoying the scenery and taking in the surrounding environment.

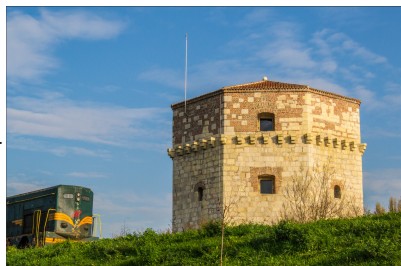

**Self-Augmented Round-1**: The image captures a scene of a stone tower standing tall against a backdrop of a clear blue sky dotted with white clouds. The tower, with its red roof and white walls, exudes an air of antiquity. It's perched on a grassy hill, adding to its grandeur. A flagpole rises from the tower, reaching towards the sky. In the foreground, a train engine painted in vibrant shades of yellow and green adds a touch of modernity to the otherwise historical setting. The perspective of the image places the viewer at a lower vantage point, looking up at the tower, enhancing its imposing presence.

**Self-Augmented Round-3**: The image captures a scene of a green train parked on a track, with a brick tower standing tall in the background. The tower, constructed from red bricks, features a pointed roof and a flagpole on top, adding a touch of historical charm to the scene. The sky above is a clear blue, dotted with white clouds, creating a serene backdrop for the tower. The perspective of the image is from the ground, looking up at the tower, giving it a sense of grandeur and prominence.

**Specialist-Augmented**: The image captures a scene of a green train car, adorned with a yellow stripe, making its way through a verdant landscape. The train car, positioned on the left side of the frame, is moving towards the right, suggesting a journey in progress. On the right side of the image, a stone tower stands tall. It's a two-story structure, topped with a red roof that contrasts with the surrounding greenery. The tower is situated on a grassy hill, adding a sense of depth to the scene. The sky above is a clear blue, dotted with white clouds, providing a serene backdrop to the scene. The train car and the tower are the main subjects of the image, their contrasting colors and shapes creating a visually interesting composition. The train car's movement adds a dynamic element to the otherwise tranquil setting. Overall, the image presents a harmonious blend of man-made structures and natural elements, each contributing to the narrative of the scene.

Figure 6: More examples of comparison among captions from generation rounds of VILA$^2$ and the widely used caption baseline of InstructBLIP (Dai et al., 2023).

**InstructBLIP**: The image depicts a crowd of people watching an old-fashioned car parade on a cobblestone street. There is a white vintage car in the foreground, while several other cars can be seen scattered throughout the scene. Some of the vehicles are placed closer to the center of the road, while others are spread out over a larger area. Overall, the scene offers a sense of nostalgia and celebration with the presence of these classic vehicles.

**Self-Augmented Round-1**: The image captures a vibrant scene on a cobblestone street. Dominating the frame is a white vintage car, its black roof contrasting with its body. The car is parked, its front facing the camera, as if ready to embark on a journey. The license plate, reading \"22-22\", is clearly visible, hinting at the car's unique identity. In the background, a yellow car adds a splash of color to the scene, parked further down the street. The cobblestone street, lined with buildings, extends into the distance, creating a sense of depth and perspective. People can be seen walking on the sidewalk, going about their day, oblivious to the vintage charm of the cars. Their presence adds a touch of everyday life to the scene, grounding the image in reality. The image is a blend of the old and the new, the vintage car standing out against the modern cityscape. It's a snapshot of a moment, frozen in time, telling a story of a city where the past and present coexist.

**Self-Augmented Round-3**: The image captures a vibrant scene on a cobblestone street. Dominating the frame is a white vintage car, its black roof contrasting with its body. The car is parked, its front facing the camera, as if ready to embark on a journey. The license plate, reading \"22-22\", is clearly visible, hinting at the car's unique identity. In the background, a yellow car adds a splash of color to the scene, parked further down the street. The cobblestone street, lined with buildings, extends into the distance, creating a sense of depth and perspective. People can be seen walking on the sidewalk, going about their day, oblivious to the vintage charm of the cars. Their presence adds a touch of everyday life to the scene, grounding the image in reality. The image is a blend of the old and the new, the vintage car standing out against the modern cityscape. It's a snapshot of a moment, frozen in time, telling a story of a city where the past and present coexist.

**Specialist-Augmented**: The image captures a scene on a city street. Dominating the frame is a white vintage car, its shiny exterior reflecting the light. The car is parked on a brick pavement, its chrome bumper and grille gleaming under the sunlight. The car's design is reminiscent of a bygone era, adding a touch of nostalgia to the modern cityscape. In the background, a yellow car can be seen, its bright color contrasting with the white of the vintage car. The yellow car is parked further down the street, ready to ferry passengers to their destinations. The street itself is lined with buildings, their architecture hinting at the urban setting. A large building with a glass facade stands out in the background, its modern design juxtaposing the vintage charm of the car. The image is taken from a low angle, looking up at the car and the buildings. This perspective gives a sense of grandeur to the scene, making the car and the buildings appear larger and more imposing. Overall, the image presents a blend of old and new, nature and architecture, stillness and motion. It's a snapshot of a moment in a city, where the past and present coexist.

Figure 7: More examples of comparison among captions from generation rounds of VILA$^2$ and the widely used caption baseline of InstructBLIP (Dai et al., 2023).

**InstructBLIP-FLAN-T5-XXL**: The image depicts a large swimming pool in the middle of a lush green landscape. There are several lounge chairs placed around the pool, and there is also a group of people lounging near the edge of the pool. A number of trees can be seen surrounding the pool, providing a picturesque backdrop for the scene.

**LLaVA-NeXT-34B**: The image depicts a serene and well-maintained swimming pool area. The pool itself is rectangular with a curved edge on one side, suggesting a design that accommodates both swimming and lounging. The water appears clear and inviting, reflecting the bright sunlight. In the foreground, there are several sun loungers, arranged in a semi-circle around the pool, inviting relaxation and sunbathing. The loungers are empty, which adds to the tranquil atmosphere of the scene. The pool is surrounded by a stone deck, which provides a solid and elegant base for the pool area. The deck's design complements the pool's shape, creating a harmonious visual flow. In the background, there is a large, well-manicured lawn that extends to a line of trees, providing a natural boundary and a sense of privacy. The lawn is lush and green, indicating that the image was likely taken during a warm season. Above the lawn, there is a large, white gazebo with a red roof, which stands out against the greenery. The gazebo's presence suggests a place for social gatherings or perhaps a spot for a quiet read. The overall spatial relationship of the elements in the image is balanced and inviting. The pool is centrally located, drawing the eye, while the sun loungers and gazebo are positioned to offer different experiences for visitors. The lawn and trees provide a sense of openness and connection to nature, while the stone deck and gazebo offer more structured spaces for relaxation. The image does not contain any people, which gives it a peaceful and undisturbed quality. The absence of human activity allows the viewer to focus on the design and layout of the pool area. The sky is clear and blue, suggesting a sunny day, which is ideal for outdoor activities. The sunlight illuminates the scene, casting soft shadows and highlighting the textures of the stone and the smoothness of the water. Overall, the image presents a luxurious and relaxing outdoor space, designed with attention to detail and a focus on comfort and aesthetics. The spatial relationships between the various elements create a harmonious and inviting environment.

**VILA^2-7B**: The image captures a serene outdoor setting, dominated by a large, light blue pool that sits in the center. The pool is encircled by a white wall, which is adorned with a series of small waterfalls, adding a soothing ambiance to the scene. On the right side of the pool, a red umbrella stands out against the natural backdrop, providing a spot of color and shade. The pool area is not devoid of life, with a few people enjoying the tranquil environment. Some are lounging on the poolside, while others are taking a dip in the water. The pool is nestled amidst lush greenery, with trees and bushes enveloping the area, creating a sense of seclusion and tranquility. The sky above is a clear blue, suggesting a bright, sunny day. The overall scene is one of relaxation and leisure, a perfect depiction of a day spent by the pool.

Figure 8: More examples of comparison among captions from generation rounds of VILA$^2$ and an additional caption baseline of LLaVA-NeXT (Liu et al., 2024c). LLaVA-NeXT-34B tends to generate longer caption with more hallucinations.

