# OpenReview forum: "VILA^2: VLM Augmented VLM with Self-Improvement"
_ICLR.cc/2025/Conference — ICLR 2025 Conference Withdrawn Submission_

### Official Review · Reviewer_SxvX · 2024-11-01

**Soundness:** 2
**Presentation:** 2
**Contribution:** 2
**Rating:** 3
**Confidence:** 5

**Summary:**

This work explores improving VLMs by self-training including a self-augment step and a specialist-augment step to iteratively improve data quality and hence, model performance. They demonstrate that synthetic data, combined with the original data, can collaboratively generate stronger models in a bootstrapped loop manner.  The specialists with a focus on new knowledge or tasks, such as a spatial-aware expert, OCR expert or grounding expert, are finetuned from the self-augmented VLM using a limited amount of additional SFT data.
The specialists can then recaption a massive amount of pretraining data.

**Strengths:**

The Figures/Tables are clear to read and understand. The problem is interesting to the community. The details are enough to reproduce the main method.

**Weaknesses:**

1. Some sentences are hard to read, e.g., Line 19-23. A proofread would be helpful.
2. The argument in Line 51-52 may not be accurate. There are emerging works [1,2,3]  about improving large-scale image-caption data, which shows it is realistic to improve the data without human annotations.
3. My biggest concern is about the effectiveness and the real benefit from the proposed approach. From Table 1, it seems the self-augment step can only bring very trivial improvements. For the specialist step, they use additional datasets, such as LV3D,  GRIT to fine-tune the model to learn spatial / grounding / OCR abilities, which may not be fair to name as "self-augment" as the model is distilling from outer source datasets. What if we directly use these data for training?
4. If the self-distillation / refinement is helpful, what about cross-model distillation? Can we leverage more public models and achieve better self-training effect?


[1] Li, Xianhang, Haoqin Tu, Mude Hui, Zeyu Wang, Bingchen Zhao, Junfei Xiao, Sucheng Ren et al. "What If We Recaption Billions of Web Images with LLaMA-3?." arXiv preprint arXiv:2406.08478 (2024).
[2] Lai, Zhengfeng, Haotian Zhang, Bowen Zhang, Wentao Wu, Haoping Bai, Aleksei Timofeev, Xianzhi Du et al. "Veclip: Improving clip training via visual-enriched captions, 2023." URL https://arxiv. org/abs/2310.07699.
[3] Chen, Lin, Jinsong Li, Xiaoyi Dong, Pan Zhang, Conghui He, Jiaqi Wang, Feng Zhao, and Dahua Lin. "Sharegpt4v: Improving large multi-modal models with better captions." arXiv preprint arXiv:2311.12793 (2023).

**Questions:**

1. There are many small typos in the paper, a proofread is needed. For example, Line 179, “concatenated" -> “concatenated”

---

### Official Review · Reviewer_fjHx · 2024-11-01

**Soundness:** 3
**Presentation:** 3
**Contribution:** 2
**Rating:** 5
**Confidence:** 4

**Summary:**

This paper introduces VILA², a novel approach to improving visual language models (VLMs) through self-augmentation of training data. The key innovation is using the VLM itself to enhance its training data quality through two main steps: (1) a self-augment loop where the VLM iteratively improves caption quality by recaptioning its training images and (2) a specialist-augment step that leverages domain-specific experts (spatial, grounding, OCR) to inject specialized knowledge. The authors demonstrate that this approach improves performance across multiple benchmarks while being 300x more cost-efficient than human labeling. The work provides detailed ablation studies showing the effectiveness of both augmentation steps and validates the quality improvements through multiple evaluation methods, including human assessment and commercial VLM judgments.

**Strengths:**

- Motivation: The authors address a critical bottleneck in VLM training (data quality) and demonstrate substantial cost savings (300x) compared to human labeling. Results show consistent improvements across diverse benchmarks and provide usable insights for this field.
- Methods: The authors propose a creative solution to the data quality bottleneck in VLM training. They innovatively use specialist models to inject domain-specific knowledge.
- Experiments: Comprehensive empirical validation through multiple benchmarks. Besides public benchmarks, they also conduct multiple evaluation approaches (human evaluation, GPT-4V/Gemini verification, left-out benchmark testing). They have also clearly demonstrated that improvements come from data quality rather than just additional computation.
- Clarity: Well-structured presentation of the methodology. The authors introduce a clear illustrations of the two-step process, which is shown in Figure 1. Furthermore, they transparently discuss the limitations of their methods.

**Weaknesses:**

While the paper presents compelling results, there are several areas that could be strengthened:
- The theoretical foundations need more development. In particular, the authors observe that self-augmentation saturates after 3 rounds, but don't provide a theoretical explanation for this phenomenon. Understanding why this saturation occurs would make the method more principled and potentially help predict optimal stopping points for different scenarios.
- The exploration of the method's robustness and limitations feels incomplete. Though the authors test their approach on several benchmarks, they don't deeply analyze failure cases or explore potential negative consequences of iterative self-improvement, such as the amplification of biases or the potential reduction in output diversity. For example, the VQAv2 benchmark, where the answers are quite short, and image contents are not that diverse. It's hard to let people believe that the long captions may benefit from the proposed data pipeline,
- The choice of specialist domains (spatial, grounding, OCR) appears somewhat arbitrary. While these domains prove beneficial, the paper would be stronger with a more systematic justification for these choices and an exploration of other potential specialist areas that might provide complementary benefits. Also, I feel a data leakage problem exists; these experts are trained with the exact datasets used for downstream evaluation. I think this is not called "knowledge distillation", but more towards something like "fitting evaluation distribution" .
- The relationship between caption quality improvements and model performance could be analyzed more rigorously. While the authors show that captions become longer and more detailed through iterations, they don't fully disentangle whether the benefits come from length, accuracy, or other aspects of the generated captions.
- Scalability issue. For larger models, limited pre-training datasets for VILA might still not be enough. Though the authors demonstrate cost savings compared to human labeling, it's still unknown when scaling up to even larger quality. This information would be valuable for practitioners looking to implement the method.

**Questions:**

VILA², compared to VILA, the major contribution is self-augmented pre-training datasets and its curation methods. I rate this paper as "marginally below the acceptance". Please turn to the weakness section for my questions. I would be very happy to give this paper a raise if the authors better explain my confusion.

**Details Of Ethics Concerns:**

The authors claim a release for their datasets upon publication. As the datasets mainly discuss the dataset curation and re-written process by LLM/VLM, there may be some slight concerns that the image contents / re-captioning may contain harmful and biased information. Also, where these images are from for the pre-training, as well as the images used for re-training their experts, might also raise the concern.

---

### Official Review · Reviewer_Gvpi · 2024-11-02

**Soundness:** 3
**Presentation:** 2
**Contribution:** 2
**Rating:** 5
**Confidence:** 3

**Summary:**

1. The self-improvement method mentioned in this paper can effectively enhance the model's performance.
2. The paper conducts comprehensive experiments to demonstrate the effectiveness of the self-augment step and the specialist-augment approach.

**Strengths:**

Originality: This paper applies the concept of self-improvement to the VLLM model (a new domain).

Quality: The paper provides a detailed analysis of self-augmentation, covering prompt templates, iteration rounds, and saturation points; it also presents a practical method that employs specialist-augmentation for continuous improvement. The experimental comparisons are comprehensive and solid.

Clarity: The descriptions of the methods and the details of the experiments are relatively clear.

Significance: Self-improvement is valuable for enhancing model performance in constrained scenarios.

**Weaknesses:**

1. Lack of Novelty: The concept of self-improvement has been extensively researched and applied, as evidenced by works such as https://arxiv.org/pdf/2202.12040, https://arxiv.org/pdf/2210.11610, and https://arxiv.org/pdf/2404.12253.
2. Limitations of the Method: While self-improvement can enhance model performance in specific scenarios, it has significant limitations, such as how to mitigate the data error accumulation caused by self-augmentation.

**Questions:**

1. Issues with Paper Presentation: In line 178 of the paper, the statement “Keeping Original Human Text is Important. We compare different conversation templates in Table 1.” seems to contain an error; it may actually refer to Table 2.
2. Necessary Comparison for Specialist-Augmenting: When validating the effect of Specialist-augmenting (relative to VILA4), as shown in Table 3, additional datasets (such as SpatialRelationQA, GRIT dataset, and the dataset for ORC specialist) were introduced to train the VLM, whereas VILA4 did not utilize these datasets. This comparison is unfair and does not adequately demonstrate the role of Specialist-augmenting.

---

### Official Review · Reviewer_VCAB · 2024-11-03

**Soundness:** 3
**Presentation:** 3
**Contribution:** 2
**Rating:** 5
**Confidence:** 5

**Summary:**

The paper proposes a self-augmentation strategy, which iteratively improves data quality via three self-augmenting loops. The authors carefully design the augmentation strategy from two aspects: self-augmentation and  specialist-augmentation. Experimental results show that the proposed method has improved performance.

**Strengths:**

1. The paper is well written and easy to follow.
2. The topic of self-improvement is interesting, which will be significant for future research.
3. Extensive experiments show that the proposed method can improve the performance.

**Weaknesses:**

1. The author's motivation is whether it is possible that the VLM itself can remedy dataset deficiency and enhance its training, but why only conduct experiments in the pre-training stage instead of the SFT stage. As far as I know, data acquisition in the pre-training stage is relatively easy and cheap, and there is no work that calls the black-block API on a large scale to obtain pre-training data (such as 50M used in the paper).
2. Line 74 mentions that ‘’Intuitively and as we observed, the loops offer performance boosts for free…’’. The method proposed in this paper requires the cost of data synthesis and longer training time, so why is it free?
3. Since the method in this paper actually uses more synthetic data, whether directly expanding the data scale can achieve similar results (such as sampling more data from coyo,laion).
4. The comparison in Table 5 is actually not fair. These methods all use less pre-training and SFT data. Besides, Table 5 lacks comparisons with the latest methods, such as InternVL1.5, MiniCPMv2.5, etc.

**Questions:**

See weakness

---

### Note · Authors · 2024-11-15

I have read and agree with the venue's withdrawal policy on behalf of myself and my co-authors.